# Hyperspectral longwave infrared reflectance spectra of naturally dried algae, anthropogenic plastics, sands and shells

Shungudzemwoyo P. Garaba [1], Tomás Acuña-Ruz [2] and Cristian B. Mattar [3]

[1]Marine Sensor Systems Group, Institute for Chemistry and Biology of the Marine Environment, Carl von Ossietzky University of Oldenburg, Schleusenstraße 1, Wilhelmshaven 26382, Germany

[2] Laboratory for Analysis of the Biosphere (LAB), University of Chile, Av. Santa Rosa 11315, La Pintana Santiago. Chile.

[3] Laboratory of Geosciences (Geolab), University of Aysén, Obispo Vielmo 62, Coyhaique, Chile.

*Correspondence to*: Shungudzemwoyo P. Garaba (shungu.garaba@uni-oldenburg.de)

## Abstract

Remote sensing of litter is foreseen to become an important source of additional information relevant to scientific awareness about plastic pollution. Here, we document directional hemispherical reflectance measurements of anthropogenic and natural materials gathered along the shorelines of Chiloé Archipelago, Chile. These spectral observations were completed in a Chilean laboratory using a state-of-the-art hyperspectral HyLogger-3™ thermal infrared (TIR) spectrometer starting from the medium wave infrared (6 µm) to long wave infrared (14.5 µm) spectrum at 0.025 µm intervals. The samples we investigated included sands, shells, algae, nautical ropes, Styrofoam®, gunny sacks and several fragments of plastic-based items. The apparent visible colours of these samples included shades of black, blue, brown, green, orange, white and yellow. We grouped the samples using robust statistical approaches (derivatives, peak seeking technique) and visual analyses of the derived hyperspectral reflectances. In each group we derived an average or TIR end-member signal as well as deduced diagnostic wavebands. Most of the diagnostic wavebands picked were found to be inside the atmospheric window of the TIR spectrum region. Furthermore, this laboratory reference dataset and findings might become useful in related field observations using similar thermal infrared technologies, especially in identifying anomalies resulting from environmental and meteorological perturbations. Validation and verification of proposed diagnostic wavebands would be part of a continuing effort to advance TIR remote sensing knowledge as well as support robust detection algorithm development to potentially distinguish plastics in litter throughout the natural environments. Data is available in open-access via the online repository PANGAEA database of the World Data Centre for Marine Environmental Sciences https://doi.pangaea.de/10.1594/PANGAEA.919536 (Acuña-Ruz and Mattar B., 2020).

## 1 Introduction

Investigations focused on remote sensing technologies with 'detect, identify, quantify, track' capabilities for floating plastic litter in the blue planet are gaining momentum (Garaba and Dierssen, 2020;Maximenko et al., 2016). Consequently, there has been an increasing amount of exploratory research gathering spectral information in the ultraviolet (UV, 0.35 µm) to shortwave

infrared (SWIR, 2.5 µm) spectrum. In these studies, multi to hyperspectral reflectance signatures of plastic litter were gathered in laboratories, over land as well as in aquatic environments from handheld, unmanned aerial systems, aircraft and satellite platforms (Garaba et al., 2018;Topouzelis et al., 2019;Acuña-Ruz et al., 2018;Goddijn-Murphy and Dufaur, 2018). Likewise, stakeholders in collaboration with remote sensing scientists have been dedicating resources to examine opportunities of

leveraging descriptor information or proxy end-products from these technologies to support monitoring of plastic litter.

An interdisciplinary team of experts recently proposed some initial requirements and capabilities for future sensors relevant to the detection of plastics in terms of temporal and geo-spatial resolutions (Martínez-Vicente et al., 2019). Despite the need for new sensors alongside the current suite, remote sensing technologies ought to offer an integrated, affordable and sustainable

environmental monitoring system (Maximenko et al., 2016;G20, 2017;Werner et al., 2016). Furthermore, there is a research gap in scientific evidence-based assessments that have utilized a wide range of current in-situ, airborne and spaceborne remote sensing tools to monitor plastic litter in the blue and green planet. Technologies with potential applications include synthetic aperture radar, polarimeters, light detection and ranging, microwave and thermal infrared imaging (Goddijn-Murphy and Williamson, 2019;van Sebille et al., 2020;Garaba et al., 2018;Lebreton et al., 2018). Thermal infrared (TIR) sensors have been

shown to be suitable and effective in collecting information about heterogeneous targets such as clouds, rare earth elements, soils, oil (hydrocarbon) spills even tracking whales or ships at sea (Laakso et al., 2019;Salisbury et al., 1987;Cuyler et al., 1992;Sobrino et al., 2009;Becker et al., 1981;Salisbury et al., 1993;Kuenzer and Dech, 2013;Hulley and Hook, 2008). The scientific evidence-based knowledge gained already from prior works on TIR remote sensing of oil, a natural hydrocarbon, can be utilized to investigate prospects of detecting floating plastic litter because plastics found in floating aquatic litter have

been identified to be mostly synthetic hydrocarbons (GESAMP, 2015;Thevenon et al., 2014).

The framework of this work is therefore to highlight hyperspectral reflectance measurements of anthropogenic and natural objects conducted in the medium wave (MWIR, 6 µm) to long wave infrared (LWIR, 14.5 µm) spectrum. We further characterize the observed spectral reflectance properties and identify distinguishing features using robust statistical techniques.

Within the scope of our study, we did not further discriminate the sand, algae and polymer types within the samples as focus was on overall identification of optical properties of plastics in marine or washed ashore litter. Our hyperspectral reference library established and explained in this study, further improves scientific knowledge about TIR characteristics of natural and anthropogenic plastic material. We believe such knowledge is essential in remote sensing algorithm development and defining future mission requirements (signal to noise, diagnostic wavebands, bandwidth, geo-spatial or spectral resolution).

## 2 Methods and Materials

### 2.1 Samples

Litter was gathered along the shorelines of Punta Mallil-Cuem, Detif and Punta Apabón on Chiloé Archipelago, Los Lagos region of Chile from January to February 2017 (Acuña-Ruz et al., 2018). Synthetic and manmade litter items included buoys, nautical ropes, fish nets, meshes, plastic bottles, bags, strapping bands, tubings, gunny sacks, Styrofoam®, ordinary ropes, placemats (**Figure 1**). Visual inspection of litter samples suggested short to long term exposure to natural weathering processes in the environment. Several objects seemed to have significant apparent variations in colour or brightness, we therefore completed measurements on the respective surfaces to assess the effects of these differences on the reflectance (**Figure 1b, c, g**). For brevity, the surfaces were identified as inside and outside for these individual objects and no cutting or other preparations were done on these materials. Qualitative analyses of the litter were focussed on the apparent colour, shape and form, factors that were used to predict the original state of the individual objects. Observed apparent colours of litter included shades of black, blue, brown, green, orange, white and yellow. Each of these items had a size of at least 0.5 m² thus falling in the category of macroplastics (diameter > 5 mm). Other samples gathered were natural materials like local sand, rocks, shells and algae or vegetation for comparison purposes. The samples collected for this experiment were assumed to represent a majority of anthropogenic plastic and natural materials found along the shorelines of Chiloé Archipelago and this was consistent with floating litter obtained from a multi-year survey of other regions in Chile (Thiel et al., 2013;Urbina et al., 2020).

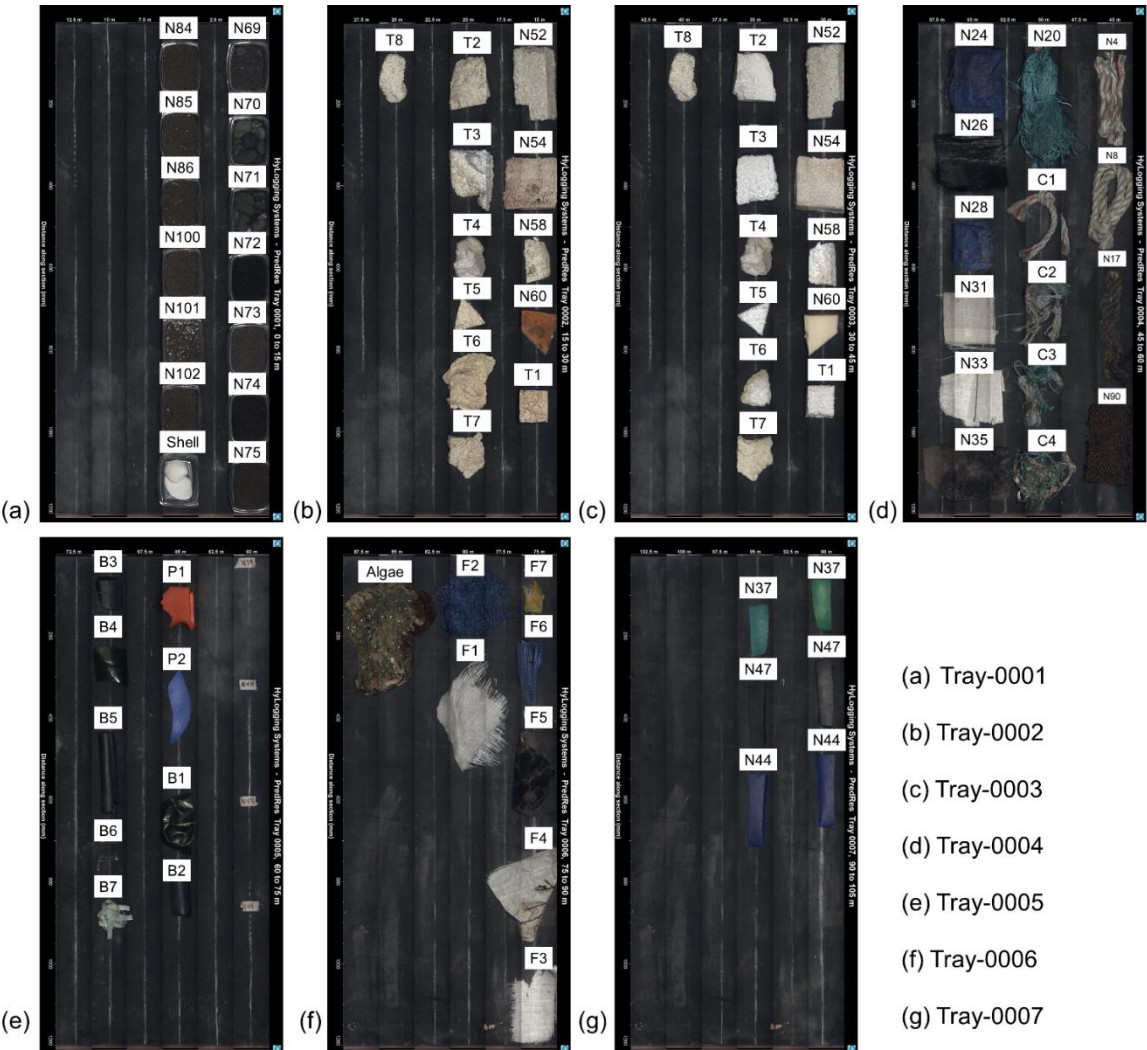

Figure 1. Naturally dried (a) sands and shells, (b-c) Styrofoam® outside and inside surfaces, (d) nautical ropes, gunny sacks and fish nets, (e) tubes, plastic bottles and buoys, (f) gunny sacks, meshes, fish nets and algae; and (g) buoys pieces collected along the shorelines of Punta Mallil-Cuem, Detif and Punta Apabón on Chiloé Archipelago, Chile from January to February 2017 placed on black trays for hyperspectral hemispherical reflectance measurements using the HyLogger-3™ spectrometer.

## 2.2 Directional hemispherical reflectance measurements

Thermal infrared spectral measurements between 6 and 14.5 µm were obtained in 0.025 µm steps using the laboratory hyperspectral HyLogger-3™ spectrometer at the University of Chile, Chile. HyLogger-3™ spectrometer has 341 wavebands and a peak signal-to-noise ratio (≥2000 at 8 µm) for a Lambertian surface with 100 % directional hemispherical spectral reflectance. Detailed specifications of the instrument have been reported in a prior study and we conducted our experiments following the proposed operating protocol of the instrument (Schodlok et al., 2016). A 10 x 10 cm Labsphere Infragold® diffuse plaque was used as a referencing standard to determine the hemispherical reflectance. Before the spectral

measurements, the algae was placed between newspapers to dry whilst the rest were left to dry naturally in the laboratory for 7 days, this step was conducted to best simulated the conditions from which the litter originated along the shoreline, a relatively dry environment exposed to wind gusts and sunlight. The dry samples were placed on black tray that was labelled before scanning. The instrument was put in automatic mode to record along-track, a measurement was recorded as an average of

continuous scans. Number of scans per object ranged between 12 to 99 scans and this was automatically set by the instrument as a function of the inherent length of each item along the track of scanning. The average object scan size was 16 mm. Detector window was at 0 ° nadir viewing angle with dual 800 °C blackbody radiation sources at 20 ° nadir angle. Altogether, 76 spectra matching the number of objects in this study were computed as the average of successive scans over each respective item. A true colour image was also captured during the scanning of each subset placed on the tray.

**2.3 Data analyses**

Processing of collected data was performed in MathWorks MATLAB R2016a. Descriptive statistics (mean, standard deviation and median) were computed from the 76 items inspected (**Figure 1**). Location of absorption features were the main descriptors of spectra, a common technique in remote sensing (Garaba and Dierssen, 2020;Dierssen and Garaba, 2020;Huguenin and Jones, 1986). Locations of absorption features were derived using a modified and robust scale-space peak algorithm (Liutkus,

2015). Instead of identifying the peaks in the measured spectra, we first inverted the spectra to transform the actual peaks into absorption features whilst making the absorption features into peaks. After peak waveband identification the measured spectra were smoothed using a moving average filter with a window of 0.003 µm, second derivatives were computed using 3-point central differences. No quantitative spectral shape evaluation was conducted because the analyses completed were considered appropriate to objectively classify and characterise the spectra available. The waveband locations of diagnostic absorption

features were first obtained from the modified scale-space peak algorithm and further confirmed through major peaks revealed in second derivative spectra. Representative end-members were estimated as the average of all spectra in each proposed group. Unbiased percentage difference (UPD) was computed to determine the variability or uncertainty of observed spectra within each group of associated materials as proposed in prior studies (Garaba and Zielinski, 2013;Garaba et al., 2015). A low UPD indicated little differences in the magnitude of the measured reflectances of items in the same group whilst a high value

suggested significant differences. The classification of these materials was therefore based on spectral characteristics revealed by the statistical analyses complemented by careful visual inspection of the objects (**Figure 1**).

## 3 Results

### 3.1 Natural materials

#### 3.1.1 Sands

A total of thirteen dark looking sand samples were gathered along the shorelines. The measured reflectances varied in terms of magnitude over the measured wavebands (UPD: mean = 118 ± 26% min = 67%, max = 170%). Sample N72 and N74 had the lowest reflectances which can be attributed to their apparent colours as the darkest targets, hence less reflective (**Figure 1a**). These sands had a generally flat signal from 6 to 7.8 µm, except for samples N101 that consisted of a mix of sand and shiny fragments of shells (**Figure 1a**). We assume these fragments resulted in the peak observed between 6.4 and 6.8 µm. A TIR edge was noted at ~7.8 µm in all the measurements of the sand samples. Highest reflectance values were identified between 9 and 10 µm, except for sample N85 at ~8.3 µm. There was a rapid decrease in reflectance from 10 to 12.7 µm followed by a gradual change up to 14.5 µm (**Figure 2**).

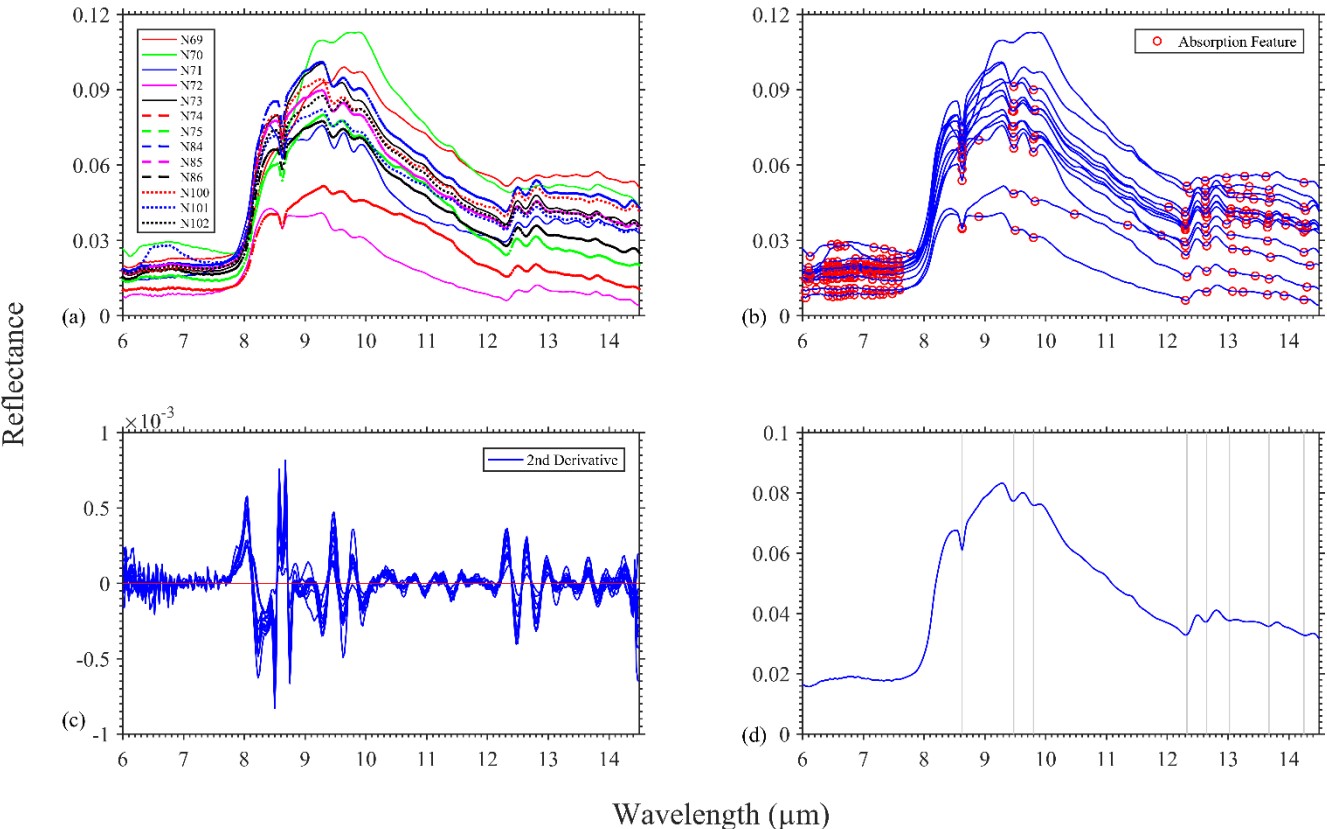

**Figure 2.** (a) Spectral reflectance of sands gathered along the shorelines of Punta Mallil-Cuem, Detif and Punta Apabón on Chiloé Archipelago, Chile, (b) absorption features automatically picked, (c) second derivatives spectra and (d) an end-member signal with the proposed diagnostic absorption features highlighted by the vertical lines.

### 3.1.2 Shells

The apparent colour of typical shells found in the Los Lagos region of Chile were off white (**Figure 1a**). Over the measured spectrum the highest reflectance was observed at ~6.8 μm, marked by the TIR edge around ~6.1 μm. Major rapid increase and decrease in reflectance were located between 6 and 7.1 μm as well as 11 and 11.8 μm, suggesting two tailing shaped peaks. Beyond the reflectance peak at ~6.8 μm, there was a gradual decrease in the measured reflectance with an exception of the peak at 11.625 μm (**Figure 3**).

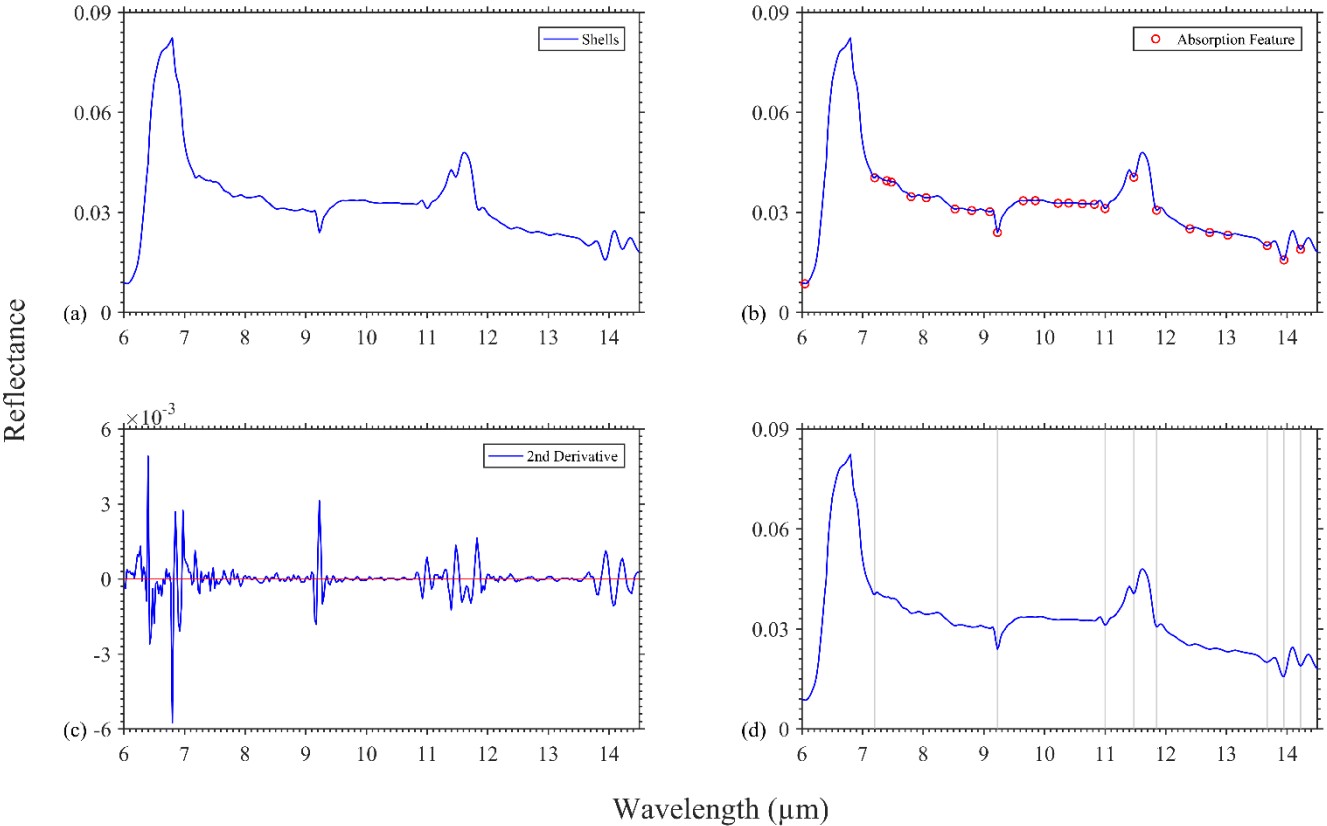

**Figure 3.** (a) Spectral reflectance of shells gathered along the shorelines of Punta Mallil-Cuem, Detif and Punta Apabón on Chiloé Archipelago, Chile, (b) absorption features automatically picked, (c) second derivatives spectra and (d) an end-member signal with the proposed diagnostic absorption features highlighted by the vertical lines.

### 3.1.3 Algae

A dark green algae sampled was sampled and it had shiny pieces (**Figure 1f**). Negative reflectances were observed in the wavebands above 12 μm, which suggested very strong absorption or detection limits of the sensor. The highest reflectance was at ~9.45 μm which was followed by a rapid decrease until 9.675 μm and then another major loss in reflectivity between 10.6 and 12 μm (**Figure 4**).

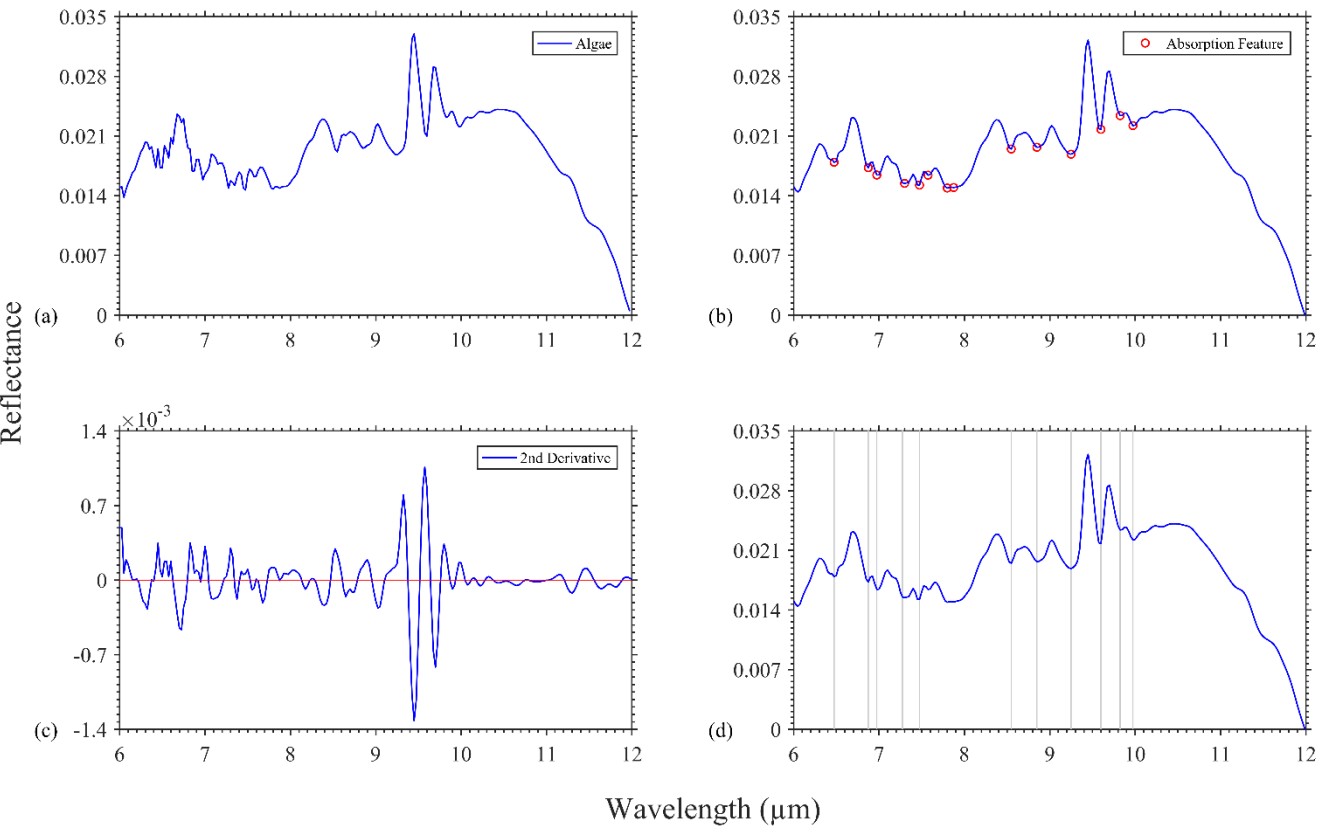

**Figure 4. (a)** (a) Spectral reflectance of algae found along the shorelines of Punta Mallil-Cuem, Detif and Punta Apabón on Chiloé Archipelago, Chile, (b) absorption features automatically picked, (c) second derivatives spectra and (d) an end-member signal with the proposed diagnostic absorption features highlighted by the vertical lines.

## 3.2 Anthropogenic materials

### 3.2.1 Styrofoam®

Spectral measurements were completed on 11 white coloured Sytrofoam® objects. Visual inspection of the top side of these pieces suggested exposure to dirt or various forms of chemical and physical weathering (**Figure 1**). The respective inside portion was relatively clean in comparison to the outside portion. The spectral shape was close to a sinusoidal curve with relatively broad peaks, narrow troughs and characterized by a slowly decreasing trend in the reflectance. At longer wavebands (>12 μm) the troughs were wider and the peaks slightly narrower relative to the wavebands below 12 μm. We noted the highest reflectances below 7 μm and decreased gradually above 14 μm. Sytrofoam® materials were the brightest targets investigated in this study and the reflectance magnitude differ over the measured wavebands (UPD: mean = 79 ± 16 %, min = 47 %, max = 120 %). Based on the determined reflectances the darkest target was upper side of N54 (**Figure 1b** and **Figure 5a**) and the brightest was bottom side of T3 (**Figure 1c** and **Figure 5a**).

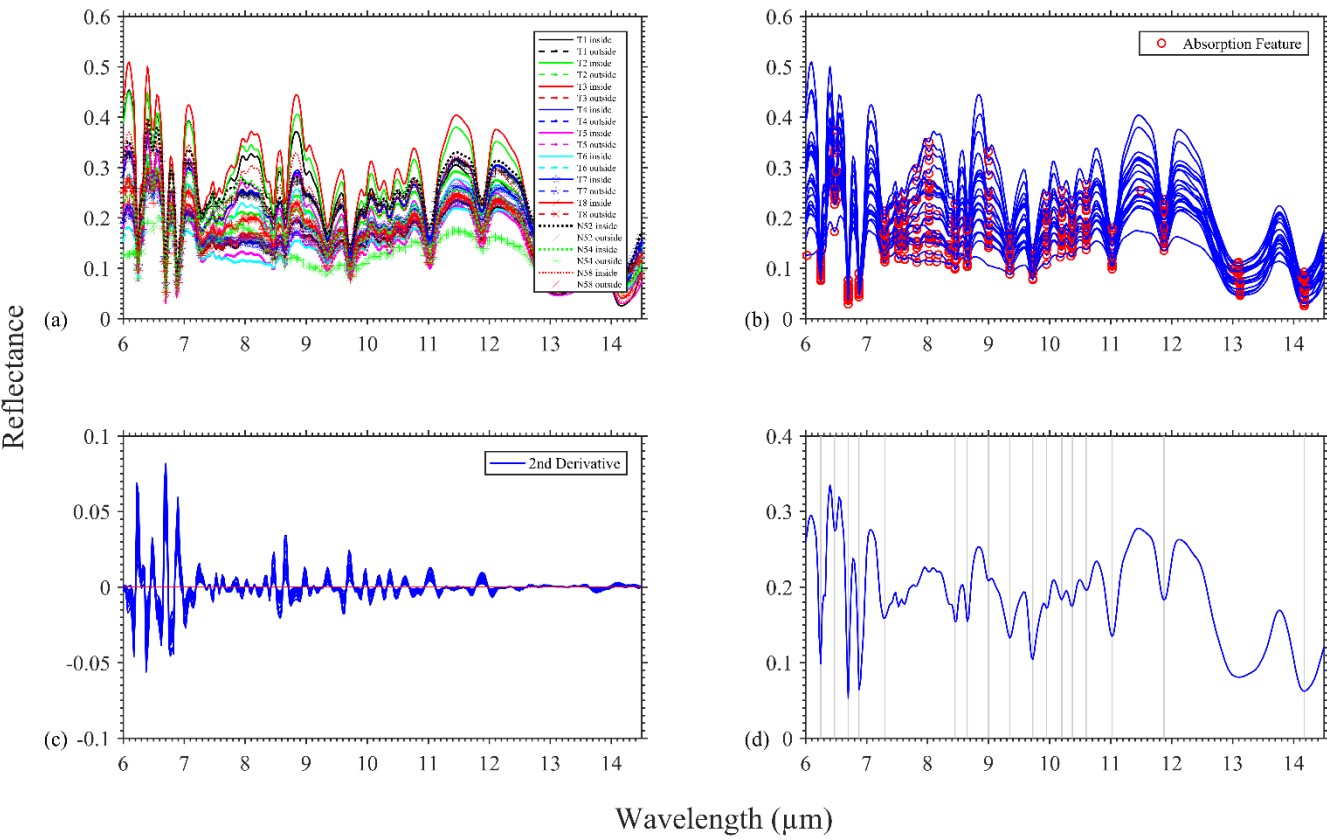

**Figure 5.** (a) Spectral reflectance of Sytrofoam® pieces found on the shorelines of Punta Mallil-Cuem, Detif and Punta Apabón on Chiloé Archipelago, Chile, (b) absorption features automatically picked, (c) second derivatives spectra and (d) an end-member signal with the proposed diagnostic absorption features highlighted by the vertical lines.

### 3.2.2 Nautical ropes and straps

Resemblances in the sinusoidal spectral shapes were salient in spectra from Sytrofoam® materials (**Figure 5**) and those of the nautical ropes as well as straps (**Figure 6**). The spectra had slightly broad peaks and narrow troughs (**Figure 6a**). We determined a generally steady increasing trend in the reflectance from the mid to the longwave infrared spectrum. Large differences in reflectance magnitude were in the longer wavebands above 13 µm (UPD: mean = 70 ± 13 %, min = 48 %, max = 101 %). On average the highest reflectances were located at ~13 µm (**Figure 6d**). The darkest target in this group was the nautical rope N17 exhibiting lowest reflectances especially for wavebands below 6.7 µm and above 10.6 µm (**Figure 1d** and **Figure 6a**).

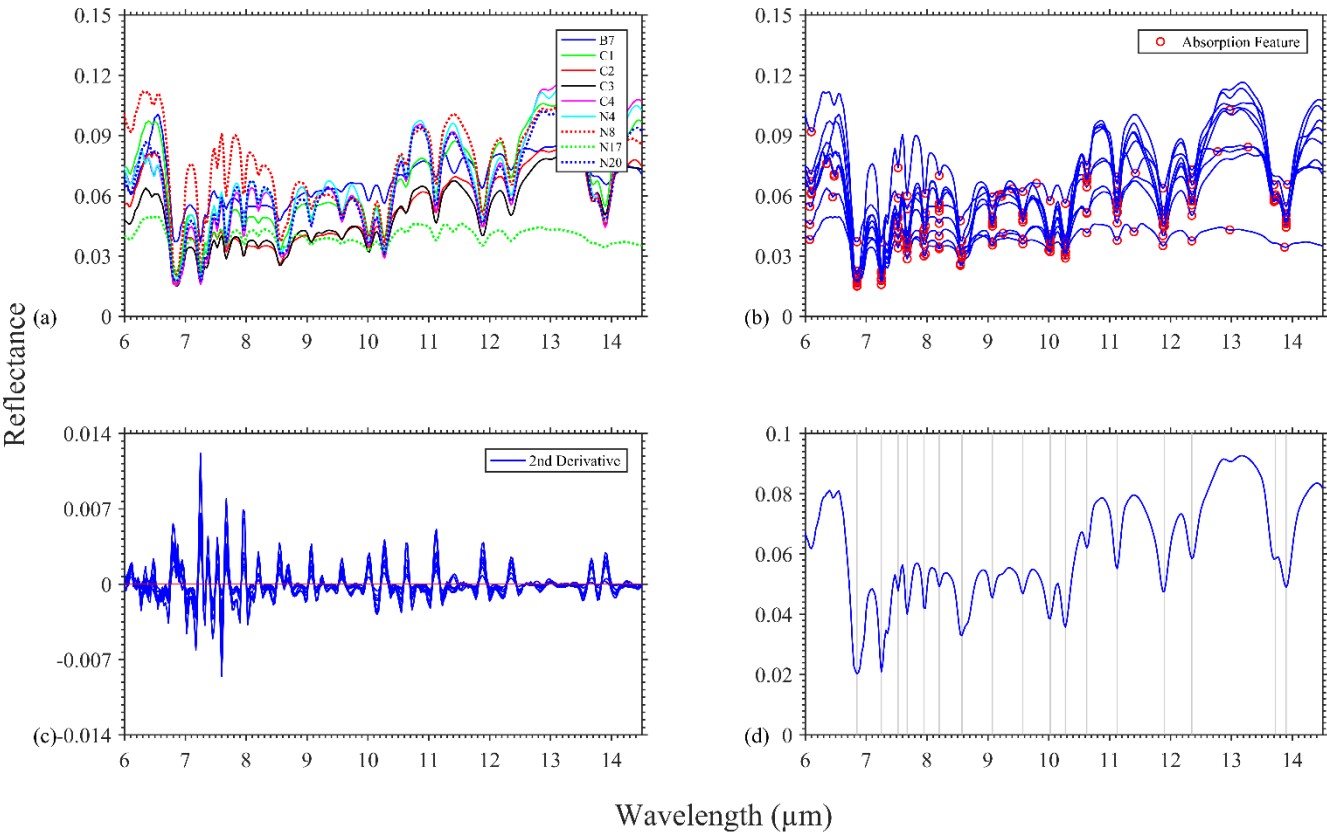

**Figure 6.** (a) Spectral reflectance of nautical ropes and straps found on the shorelines of Punta Mallil-Cuem, Detif and Punta Apabón in Chiloé Archipelago, Chile, (b) absorption features automatically picked, (c) second derivatives spectra and (d) an end-member signal with the proposed diagnostic absorption features highlighted by the vertical lines.

### 3.2.3 Gunny sacks

The apparent colours of the sacks were shades of white (F1, F3, F4, N31, N33), blue (F6) and black (N26). These whitish objects had higher reflectances compared to the darker objects within the sampled gunny sacks class (**Figure 1d**, **Figure 1f** and **Figure 7a**). Variations in the magnitude of spectra were determined to be highest below 6.8 µm and longer wavebands above 14 µm (UPD: mean = 88 ± 17 %, min = 58 %, max = 133 %). Compared to the narrow troughs, the peaks were wider and there was a moderate increasing trend in reflectance from the mid to longwave infrared.

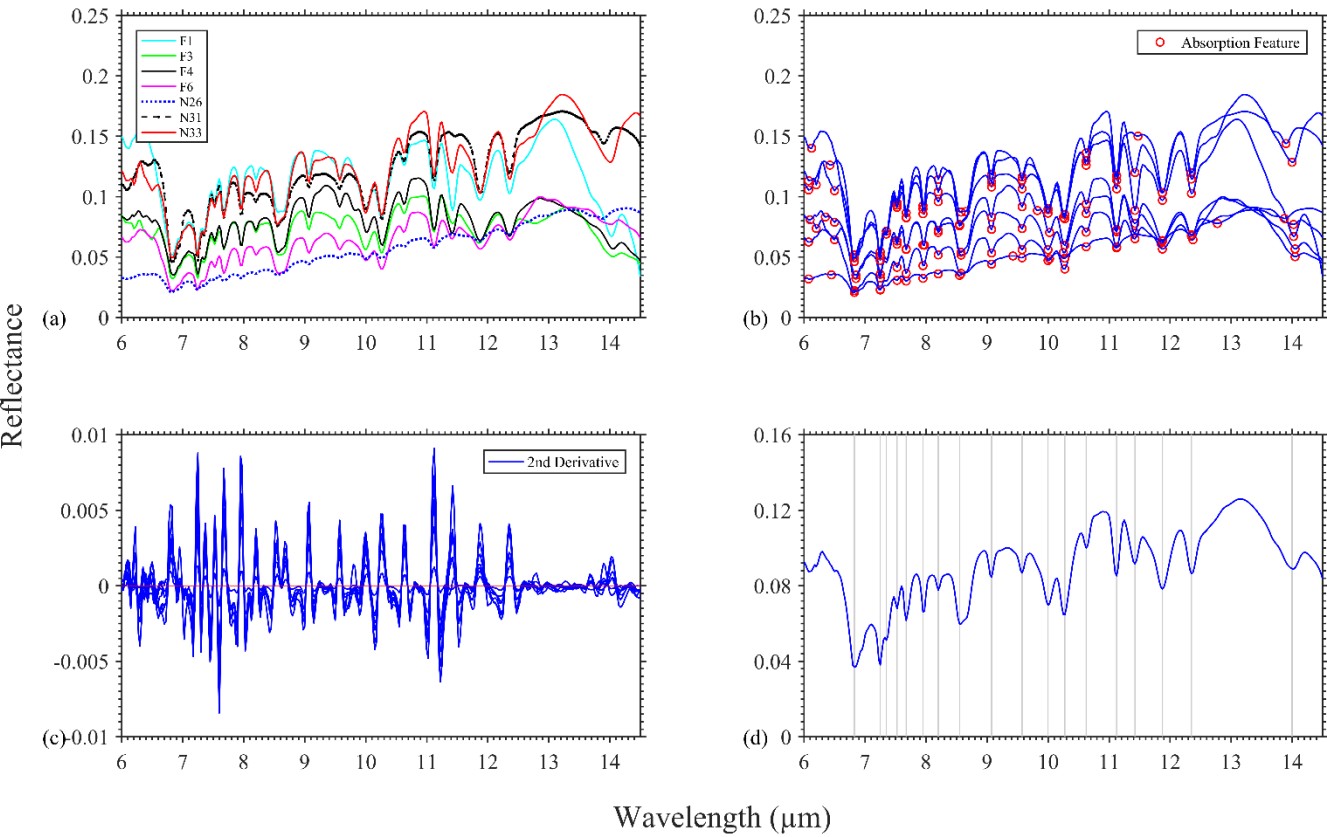

**Figure 7.** (a) Spectral reflectance of gunny sacks found along the shorelines of Punta Mallil-Cuem, Detif and Punta Apabón in Chiloé Archipelago, Chile, (b) absorption features automatically picked, (c) second derivatives spectra and (d) an end-member signal with the proposed diagnostic absorption features highlighted by the vertical lines.

### 3.2.4 Fragments

The apparent colours of collected plastic fragments varied in shades of orange (P1), blue (N44, P2), green (N37) and black (B2, B3, B5, N44). We presumed they had undergone natural weathering based on visual inspection indicated by the scratches and loss of colour (**Figure 1e** and **Figure 1g**). Interestingly, the darker targets N44 and B5 were revealed to be more reflective relative to the slightly brighter fragment, the blue P2 (**Figure 1e** and **Figure 8a**). The measured reflectances of the fragments exhibited considerable increasing trends from the mid to the longwave infrared spectrum (**Figure 8**). However, there were large margins between the less and most reflective samples (UPD: mean = 121 ± 5 %, min = 108 %, max = 129 %). On average the lowest reflectance was observed at ~6.8 μm and the highest at 14 μm, indicating a significant increasing trend (**Figure 8d**). V-shaped troughs were found at ~6.8 μm followed by U-shaped troughs between 13.5 and 14 μm, a wide plateau was identified at ~9.8 μm as well as peaks that had near rounded and tailing shapes.

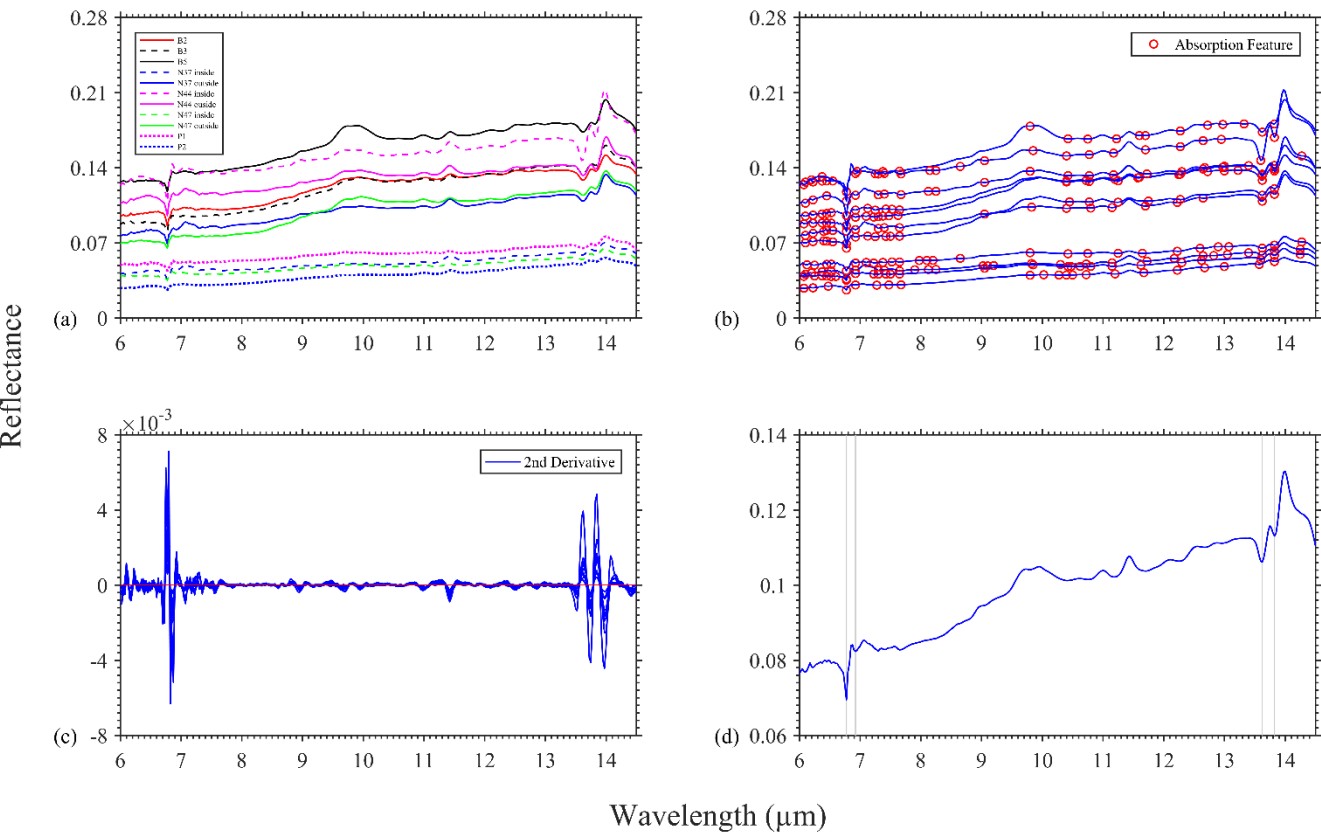

**Figure 8.** (a) Spectral reflectance of fragmented plastic objects found along the shorelines of Punta Mallil-Cuem, Detif and Punta Apabón on Chiloé Archipelago, Chile, (b) absorption features automatically picked, (c) second derivatives spectra and (d) an end-member signal with the proposed diagnostic absorption features highlighted by the vertical lines.

### 3.2.5 Plastic bottles, nets, foam and other plastics

Objects B1 and B4, fragments of green plastic bottles and a transparent strip, shared a similar spectral shape that had four major tailing peaks marked by a steady increasing trend from MWIR to LWIR (**Figure 9a**). The highest reflectance was found in the shiny green item B4, followed by the less shiny object B1 and the lowest was in the transparent strip B6 (**Figure 1e**) with large variations (UPD: mean = 89 ± 11 %, min = 59 %, max = 113 %). A blue (F2) mesh net and brownish (F7) fishing net piece were determined to have nearly rounded peaks (**Figure 1f**). Between 9 and 13 μm the range of reflectance was less than 20 % but increased rapidly from 13.5 μm to reach 194 % (UPD: mean = 30 ± 39 %, min = 0 %, max = 194 %). Around 12 μm we observed the highest reflectance and in general there was a gradual increasing trend in the signal (**Figure 9b**). Black coloured construction materials F5 (**Figure 1f**) and N35 (**Figure 1d**) showed a significant rising trend in reflectance and generally well-rounded peaks with some tailing as well as fronting peaks (**Figure 9c**). The range was moderate for these black materials (UPD: mean = 39 ± 8 %, min = 23 %, max = 61 %). Blue patches of gunny sack like material N24 and N28 (**Figure 1d**) exhibited narrow to u-shaped troughs and sharp round or symmetric peaks (**Figure 9d**). There was a gradual increasing

trend in the reflectance and it had a small range (UPD: mean = 4 ± 3 %, min = 0 %, max = 13 %). Two sides of a single white presumed polyurethane foam piece N60 (**Figure 1b** and **Figure 1c**) had a strong increasing trend in reflectance from the MWIR to LWIR (**Figure 9e**). The bright whitish side was more reflective **(Figure 1c)** compared to the dark brownish portion (**Figure 1b**), variations were also small (UPD: mean = 13 ± 7 %, min = 3 %, max = 44 %). These differences were higher in the MWIR and lowest towards the LWIR, characterised by tailing, rounded and fronting peaks as well as some narrow troughs (**Figure 9e**). A dark piece of a nautical cargo strap N90 (**Figure 1d**) was not very reflective compared to all the materials observed in this study from the MWIR to LWIR spectrum (**Figure 9f**). The measured reflectance peaked around 11.6 µm and the signal had a gradual increasing trend. Most of the peaks were rounded with some minor fronting and tailing shapes.

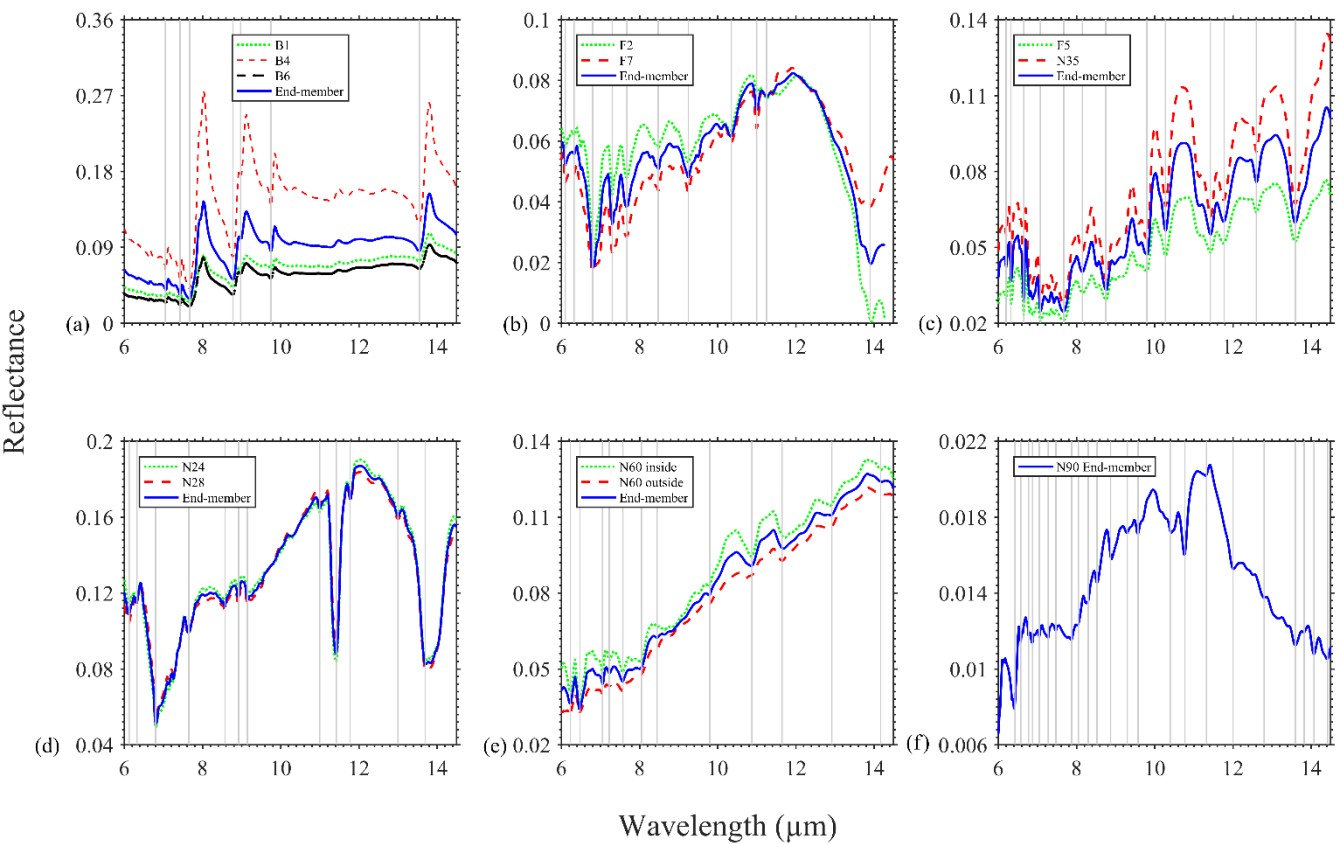

**Figure 9.** Spectral reflectance of other anthropogenic materials (a) green plastic bottle pieces and transparent strip, (b) mesh and fishing net, (c) black construction materials, (d) gunny sack like patch, (e) polyurethane foam piece and (f) nautical cargo strap in **Figure 1** gathered along the shorelines of Punta Mallil-Cuem, Detif and Punta Apabón on Chiloé Archipelago, Chile. An end-member spectrum is proposed with diagnostic absorption features highlighted by the vertical lines.

### 3.3 Diagnostic absorption features

We identified diagnostic absorption features of the measured materials based on a combination of visual inspection, modified robust scale-space peak algorithm and second derivative analyses (**Table 1**).

## 4 Discussion

The samples we investigated were presumed to be a reasonable or appropriate subset of material commonly found along the shorelines of Chiloé Archipelago, Chile. Specifically, Styrofoam®, gunny sacks and nautical ropes or the broken down fragments are typical waste products of blue economic activities, including the vast aquaculture farming in the waters of Chiloé (FAO, 2005;Thiel et al., 2013;Pozo et al., 2019;Gómez et al., 2020). However, it is important to note that our subset does not necessarily represent each and every anthropogenic and natural litter item found on Chiloé Archipelago. Global reports have already shown that there is a huge diversity in polymers, colours, sizes and shapes of plastic found in marine litter (GESAMP, 2015;Lebreton et al., 2018;Thevenon et al., 2014;Thiel et al., 2013). We are convinced our TIR sample subset provides invaluable complementary insights to the interdisciplinary scientific evidence-based knowledge of global plastic litter. To this end, it is recommended that within the TIR remote sensing community a comprehensive high quality assured and quality controlled spectral reference library be established to carefully harmonize available TIR measurements from various works e.g. ECOSTRESS (Meerdink et al., 2019), SLUM (Kotthaus et al., 2014) or contaminated anthropogenic surfaces (Kerekes et al., 2008).

**Table 1.** Location of proposed diagnostic absorption features derived from spectral measurements of samples from the shorelines of Punta Mallil-Cuem, Detif and Punta Apabón on Chiloé Archipelago, Chile.

| Material | Location of Absorption Features (µm) |
|---|---|
| Sands | 8.625, 9.475, 9.800, 12.325, 12.650, 13.025, 13.675, 14.250 |
| Shells | 7.200, 9.225, 11.000, 11.475, 11.850, 13.675, 13.950, 14.225 |
| Algae | 6.475, 6.875, 6.975, 7.275, 7.475, 8.550, 8.850, 9.250, 9.600, 9.825, 9.975 |
| Styrofoam® | 6.250, 6.475, 6.700, 6.875, 7.300, 8.450, 8.650, 9.000, 9.350, 9.725, 9.950, 10.200, 10.375, 10.600, 11.025, 11.875, 14.175 |
| Nautical Ropes and Straps | 6.850, 7.250, 7.525, 7.675, 7.950, 8.200, 8.575, 9.075, 9.575, 10.025, 10.275, 10.625, 11.125, 11.900, 12.350, 13.725, 13.900 |
| Gunny Sacks | 6.825, 7.250, 7.350, 7.525, 7.675, 7.950, 8.200, 8.550, 9.075, 9.575, 10.000, 10.275, 10.625, 11.125, 11.425, 11.875, 12.350, 14.000 |
| Fragments | 6.775, 6.925, 13.625, 13.825 |
| Plastic bottles, nets, foam and other plastics | - 7.050, 7.425, 7.675, 8.775, 8.975, 9.750, 13.550;<br>- 6.100, 6.325, 6.800, 7.300, 7.675, 8.475, 9.250, 10.350, 11.000, 11.250, 13.900;<br>- 6.200, 6.325, 6.650, 7.075, 7.675, 8.150, 8.750, 9.800, 10.275, 11.425 11.775, 12.600, 13.600;<br>- 6.125, 6.325, 6.800, 7.650, 8.575, 8.925, 9.150, 11.000, 11.425, 11.775, 13.000, 13.700;<br>- 6.225, 6.475, 7.050, 7.225, 7.575, 8.050, 8.450, 9.800, 10.875, 11.650, 12.925, 14.175;<br>- 6.425, 6.575, 6.775, 6.875, 7.050, 7.275, 7.475, 7.875, 8.050, 8.300, 8.525, 8.875, 9.300, 9.575, 10.400, 10.775, 11.325, 12.000, 12.800, 13.600, 13.825, 14.075, 14.425 |

Although, no further analyses to include the soil types, algae types and source polymer descriptors were made, we managed to identify and propose several diagnostic absorption features as a future key spectral descriptor (**Table 1**). Attaching additional physical and chemical key descriptors would add value to our spectral measurements, it is likely to improve the end-products that can be derived from remote sensing tools. The UPD metric we investigated showed how large or small the magnitude differed in our proposed groups. In general, we observed moderate percentage differences, suggesting algorithm development

based on the magnitude of reflectance might have large uncertainties. Alternatively, band ratioing algorithms or spectral shape based algorithms would mitigate the problems linked to variations in magnitude. Furthermore, we are aware that remote sensing activities in the TIR might report observations in terms of emissivity instead of reflectance (Sobrino et al., 2009;Cuyler et al., 1992). It does not present any practical limitations in terms of usage of our datasets because emissivity can be derived

from hemispherical reflectance measurements by applying Kirchhoff's Law (Nicodemus, 1965;Salisbury et al., 1987).

In this work, measurements were completed in a controlled laboratory setup and therefore did not take into account the effects of environmental and meteorological perturbations that play a role in the TIR spectrum especially in natural conditions. To this end, aquatic or land based feasibility research is required to decipher and identify scientific evidence-based capabilities of

TIR remote sensing as a monitoring strategy for plastics in land or marine litter. During our experiments the samples were left to dry naturally but future studies are urged to investigate wet samples to better simulate aquatic floating material as surface moisture has been shown to affect the detectable signal in the SWIR (Garaba and Dierssen, 2018) as well as in LWIR (Hulley et al., 2010).

Most of the diagnostic absorption features from the proposed end-member spectra were located in the spectral region where the atmosphere is transparent, also known as the atmospheric window, ranging from ~7.4 to ~14.2 μm (**Table 1**). A major benefit of these anthropogenic materials possessing diagnostic absorption features in the TIR atmospheric window for remote sensing applications would be the prospects of mitigating uncertainties or masking of signature optical properties introduced by atmospheric correction approaches. For example, if the laboratory TIR measurements presented here were to be assumed

to be in-situ Bottom-of-Atmosphere reflectances, we would potentially be able to detect the diagnostic wavebands (**Table 1**) through an intervening atmosphere, suggesting Top-of-Atmosphere reflectances (sharing similar spectral shapes with comparable magnitude). Validation and verification of this assumption would of course require radiative transfer simulations and in-situ spectral reference libraries. Furthermore, these in-situ measurements and radiative transfer simulations should be synergized with data from airborne as well as satellite platforms. The benefit of conducting aerial field surveys using multi to

hyperspectral sensors (e.g. NASA HyTES, Specim OWL, ITRES TASI-600 or SensyTech AHS) on aircrafts will be the possibility of capturing high geo-spatial TIR imagery in near-similar approaches/conditions to those of satellites i.e. an intervening atmosphere and a mobile platform. In spite of the challenges associated with varying geo-spatial resolution of remotely sensed imagery, including decreased chances to detect plastic litter in the visible spectrum (Acuña-Ruz et al., 2018), satellites provide essential information about the environment. Satellite missions with TIR sensors include ASTER from the

National Aeronautics and Space Administration/Japanese Ministry of Economy Trade and Industry, ECOSTRESS from National Aeronautics and Space Administration as well as Landsat-8 from the United States Geologic Survey. The capabilities (TIR spectral, geo-spatial, revisit interval) of ASTER, ECOSTRESS and Landsat-8 missions must be assessed with a focus on detecting aggregated litter zones, considering the geo-spatial resolutions of these sensors (38 - 100 m). We need to further emphasize that the atmospheric window in the TIR is relatively wide. This atmospheric window contained a significant number

of diagnostic wavebands of anthropogenic materials we studied and it would be vital to explore development of detection algorithms using the limited (2 - 5 wavebands) spectral information available on these current TIR missions. The TIR atmospheric window seems to offer more opportunities to better remotely sense anthropogenic plastics through an intervening atmosphere in comparison to those reported from studies covering ultraviolet to shortwave infrared spectral regions (Garaba 5 and Dierssen, 2018).

## 4 Data availability

Quality control was performed according to the guidelines of SeaDataNet. Data is available in open-access via the online repository PANGAEA database of the World Data Centre for Marine Environmental Sciences https://doi.pangaea.de/10.1594/PANGAEA.919536 (Acuña-Ruz and Mattar B., 2020).

10 ## 5 Conclusions and outlook

We report laboratory measurements of hyperspectral TIR hemispherical reflectances collected from natural and anthropogenic material. Remote sensing relevant to plastic litter in the TIR spectrum is gradually gaining research interest but not as fast as the studies based on the ultraviolet to shortwave infrared wavebands. A major drawback has been that technological advances are mostly driven by societal and scientific needs to understand the blue and green planet. Here we exhibit a spectral reference 15 library that could be leveraged to exploit and investigate the feasibility of airborne, high altitude pseudo satellite or space hyperspectral TIR sensors as additional remote monitoring strategies of plastic litter because of the wide atmospheric window, meaning Top-of-Atmosphere derived variables will not require atmospheric correction. Use of datasets that do not require atmospheric correction mitigates the uncertainties related to the various assumptions implemented in different algorithms to remove contributions by atmospheric gases among other optically active components detected by remote sensing tools. 20 Substantial work is thus required to further explore the prospects of TIR technologies as additional monitoring strategies to assist plastic litter research. Knowledge gained from studies involving plastic litter will benefit interdisciplinary science as more essential (biogeochemical and physical) proxy descriptors might be derived from current and future TIR satellite missions by agencies like European Space Agency, German Aerospace Centre, National Aeronautics and Space Administration and United States Geologic Survey.

25 **Author contribution**

SPG analysed the data and prepared the first draft of the manuscript. TAR and CBM designed and conducted the experiment. All authors discussed and approved the manuscript text.

**Competing interests**

The authors declare that they have no conflict of interest.

**Financial support**

The study was funded by Deutsche Forschungsgemeinschaft (grant no. 417276871), Discovery Element of the European Space Agency's Basic Activities (ESA contract no. 4000132037/20/NL/GLC), "Estudio para la generación de un modelo predictivo de residuos en 3 playas de Chiloé, mediante Teledetección Cuantitativa (PRED-RES Chiloé)", Agencia de Sustentabilidad y Cambio Climático (ASCC)/Sustainability Agency and Climate Change and the INNOVA CSIRO−CHILE 10CEII−9007 project (run by the HL3).

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
