# Peer review of "Hyperspectral longwave infrared reflectance spectra of naturally dried algae, anthropogenic plastics, sands and shells"

_Earth System Science Data, 2020_

## Referee Comment (RC1) · Vincenzo Palleschi (Referee) · 15 Aug 2020

The manuscript submitted reports an interesting study on the TIR reflection behavior of different kind of materials (natural and anthropogenic) collected on the shorelines of Chiloé Archipelago, Chile. The data reported are available in the online PANGEA database. The work, with the limitations discussed by the authors (the measurements were performed in a laboratory, and the materials were distinguished very coarsely between Sands, Shells and Algae, as natural materials, and Styrofoam, Nautical ropes and straps, Gunny sacks, Fragments and other Plastics) could be important for remote sensing of plastics and would surely foster further studies in this specific spectral range.

---

## Referee Comment (RC2) · Anonymous Referee #2 · 17 Aug 2020

This manuscript describes an interesting database of reflectance/emissivity spectra of some manmade plastics and natural materials. Because laboratory measurements include a wide spectral range (UV-VNIR-SWIR-TIR), this dataset can be used for identification of plastics with a number of remote sensors. The manuscript is well organized, and description of dataset is clear (and it is also freely available in PANGEA).

I have some minor suggestions to the authors:

Section 2.2: the measurement protocol is a key factor in this research. I think the section could be expanded by providing more info on the instrument (e.g. Signal-to-noise ratio for each port, etc.)

[Figure]

Section 4: Not sure if Figure 10 is really necessary at this point. However, I think it would be very interesting to discuss the limitation of current EO sensors for detection of plastics. The authors identified characteristic peaks of each reflectance spectra, so it would be interesting to discuss if these characteristic peaks can be "captured" by current EO sensors. This is probably more difficult with current TIR sensors, with limitations on the number of spectral bands.

The ECOSTRESS spectral library includes a number of manmade and natural materials. I don't know if it includes materials similar to the samples collected and measured by the authors. In this case, a rough comparison between these measurements could be an option for a brief test of the data presented by the authors.

---

## Referee Comment (RC3) · Anonymous Referee #3 · 17 Aug 2020

This manuscript describes a dataset of reflectance/emissivity spectra of dry manmade and natural materials from 6000 nm to 14000 nm. The dataset has the potential to be very useful for identification of litter in marine and coastal environments as there doesn't seem any other plastic spectral libraries available (nor even plastics in other spectral libraries such as the ECOSTRESS spectral library) and research into marine and plastic pollution is clearly gaining interest and awareness.

In general, I think the manuscript is organised well and a promising accompaniment to the dataset. However, I think a few revisions are needed before this should be published, primarily to the methodology section in order to enable maximum clarity for

users. I've gone into some details about what I think should be addressed – apologies for the length but I think it's because the dataset has the potential to be really useful.

Since this is to accompany a dataset, it needs to be very clear for the reader how the samples were collected, prepared and measured. I think it's clear from the manuscript how the samples were collected but there is limited information about preparation and further clarity is required about the sample measurement, particularly since there isn't any data available on accuracy of the HyLogger. Questions I might want to know if I were to use this dataset which are absent from the methodology include:

i. How long was there between collection and measurement?

ii. How and why were there samples dried? This is important as surface moisture has been shown to impact surface reflectance in the LWIR (e.g. see https://doi.org/10.1016/j.rse.2010.02.002). Surely wet samples might be more representative of the conditions you'd see in marine environments?

iii. What is this 'inside' and 'outside' that is mentioned in the results – did you cut into the samples to measure the 'inside'? You need to describe this since the impact this has on reflectance/emissivity is non-negligible.

iv. Was a background radiance measurement made as detailed in Schodlok et al 2016?

v. Is there any information about signal-to-noise for the instrument and the measurements?

vi. You discuss spectra being grouped into associated materials (l.27, p.4) – what are these group and how were these groups determined? Do you just mean e.g. all sands, all styrofoams?

vii. The authors refer to 'length' in line 10, p.4. What does this refer to, length of the tray or length of time?

viii. Where were the measurements made? Were the samples sent to CSRIO Australia

for measurement on the setup detailed in Schodlok et al (2016) or is there a setup in Chile where they were measured? If measurements were made using a different setup to the one in Schodlok et al (2016), I would suggest you include some more information about it and perhaps an example image of the setup during a measurement for the user. You could also perhaps could include table to present number of scans by sample/tray of samples which would be useful for the user.

ix. How did you get the spectra from the HyLogger imagery? Are your spectra the average of multiple spatial pixels?

If the samples and measurement protocol presented in this paper are the same considered in Acuña-Ruz et al. (2018), the authors could answer some of the above simply by referencing that. However, I'm not sure they are since this paper talks about 76 samples while the Acuña-Ruz et al (2018) paper talks about over 144 samples.

In terms of accompanying figures and tables, generally these are good although I think a few of the figures could benefit from further explanation in the captions. In Figure 1 for example, I think the sample key needs to explained in more detail in the caption. Also, which of the pictured repeats for N37, N44 and N47 are the 'inside' and 'outside'?

The results and discussion are in general well presented with good consistency for each subsection in the results. A couple of points I had here:

- Is the end-member presented in each subsection the mean spectra of multiple scans? Unclear from the text

- I think you would benefit from further discussion of UPD and variability as it's not clear why you have considered this nor how you have used it. If you're using it to be a measure of how trustworthy the spectra is (as I think you are?), a comparison of the different UPDs would be useful to see in the discussion

- I don't think Figure 10 is necessary

- I was surprised to see no discussion of other spectral libraries (e.g. ECOSTRESS

spectral library, SLUM spectral library) in the introduction and/or discussion given that this dataset will have a complementary role to these. If possible I would suggest you show an inter-comparison with data from these spectral libraries or other papers to help the user understand the comparable performance of your dataset since you don't have calibration or accuracy information for the HyLogger. This is especially important as you are observing unrealistic negative reflectances which could suggest inaccurate measurements. As noted earlier however, I couldn't find any plastics in the ECOSTRESS spectral library so you'd probably have to do this comparison with the sand, styrofoam or algae samples if you could find similar samples.

In terms of usefulness of this dataset, there are two points I wanted to make:

1) You identify in your discussion that a limitation of your dataset is that you don't have information on the chemical composition of your samples. However, I think using terms like 'other plastics' is very vague and will limit the use in applications – could you be at all more specific? For example, 'other plastics' seems to have multiple absorption lines, which one will users know to use? Also, in the accompanying sample pictures, are these the 'buoy samples' and 'buoy2_samples'?

2) If you're advising the user that this dataset can be used with TIR satellite sensors, you really need to address the issue of spatial and spectral resolution. Would ASTER or Landsat 8's spatial resolution really be high enough to detect samples of this kind? Even the highest resolution TIR sensors (ECOSTRESS, HyspIRI e.g.) have spatial resolutions of 60m + and with SLSTR you're looking at 1 km. If you're going to argue that this dataset can be used for satellite sensors, you'll need something similar to the discussion in e.g https://doi.org/10.1038/s41598-020-62298-z to show suitability of thermal sensors for plastic detection in oceans (and therefore why spectral library is required). If the plastic observed is < 60m, I would advise instead moving the introduction and discussion a bit more towards hyperspectral airborne TIR remote sensing (e.g. using NASA's HyTES, Specim's OWL, TASI) and thermal UAVs. Use of hyperspectral airborne sensors has the benefit of avoiding the issue of absorption features

being outside satellite spectral bands.

Regarding the dataset itself, it's accessible and easy to use (although note that the KML file is not mentioned in the accompanying publication). You could consider separating the metadata and the data for ease of use. I would also advise including a key with the sample images. The abstract here could benefit from copy-editing.

Finally, the manuscript was in general well-written but there are a few typos and incomplete sentences in the manuscript that suggest the need for a copy edit. A few I noticed in the manuscript:

1. p.14 line 7 has missing end to sentence

2. line 7 on p. 3 incorrectly says 'Were believe'

3. line 15 p.6, should this be 12000 nm rather than 1200 nm?

4. The sentence commencing l.23 on p.4: 'An inter-comparison of...' needs to be rewritten

Also, a very minor point but I would consider changing the units from nanometre to micrometre throughout as the thermal infrared spectroscopy community tends to use microns more.

---

## Author Comment (AC1) · 3 Sep 2020

Reply to reviewer Manuscript Title : Hyperspectral longwave infrared reflectance spectra of dry anthropogenic plastics and natural materials Authors : Garaba, Acuña-Ruz and B. Mattar Journal : Earth System Science Data (ESSD)

Vincenzo Palleschi (Referee)

Referee comment - 1 General Comments The manuscript submitted reports an interesting study on the TIR reflection behaviour of different kind of materials (natural and anthropogenic) collected on the shorelines of Chiloé Archipelago, Chile. The data re-

ported are available in the online PANGEA database. The work, with the limitations discussed by the authors (the measurements were performed in a laboratory, and the materials were distinguished very coarsely between Sands, Shells and Algae, as natural materials, and Styrofoam, Nautical ropes and straps, Gunny sacks, Fragments and other Plastics) could be important for remote sensing of plastics and would surely foster further studies in this specific spectral range.

Author response - 1 We appreciate your kind words and taking the time to review our manuscript.

---

## Author Comment (AC2) · 3 Sep 2020

**Reply to reviewer**

**Manuscript Title** : Hyperspectral longwave infrared reflectance spectra of dry anthropogenic plastics and natural materials
**Authors** : Garaba, Acuña-Ruz and B. Mattar
**Journal** : *Earth System Science Data* (*ESSD*)

**Anonymous Referee #2**

| Comment | Response | Revision Implemented |
|---|---|---|
| **C1**.
This manuscript describes an interesting database of reflectance/emissivity spectra of some manmade plastics and natural materials. Because laboratory measurements include a wide spectral range (UV-VNIR-SWIR-TIR), this dataset can be used for identification of plastics with a number of remote sensors. The manuscript is well organized, and description of dataset is clear (and it is also freely available in PANGEA). | **R1.**
We appreciate the positive feedback and the time taken to review our manuscript. | None |
| **C2.**
I have some minor suggestions to the authors: Section 2.2: the measurement protocol is a key factor in this research. I think the section could be expanded by providing more info on the instrument (e.g. Signal-to-noise ratio for each port, etc.) | **R2.**
Thank you for pointing this out. To the best of our knowledge, there are eight HyLogger-3™ spectrometers in the world, which have the same instrument specifications for reflectance measurements. We use the same protocols described in Schodlock et al., (2016).

We have appended this information about the instrument in the methods section to clarify this point. | *(See Section 2.2 Directional hemispherical reflectance measurements Page 4 Line 8 of the revised manuscript).* HyLogger-3™ spectrometer has 341 wavebands and a peak signal-to-noise ratio (≥2000 at 8µm) for a Lambertian surface with 100 % directional hemispherical spectral reflectance. Detailed specifications of the instrument have been reported in a prior study and we conducted our experiments following the proposed operating protocol of the instrument (Schodlok et al., 2016). |
| **C3**.
Section 4: Not sure if Figure 10 is really necessary at this point. | **R3.**
We agree with the reviewer. | Figure 10 has been removed. |

| Comment | Response | Revision Implemented |
|---|---|---|
| **C4.**
However, I think it would be very interesting to discuss the limitation of current EO sensors for detection of plastics. The authors identified characteristic peaks of each reflectance spectra,
so it would be interesting to discuss if these characteristic peaks can be "captured"
by current EO sensors. This is probably more difficult with current TIR sensors, with
limitations on the number of spectral bands. | **R4.**
This is a good point. The current TIR missions have moderate geo-spatial and limited spectral resolutions but this can be resolved by utilizing airborne or shipborne platforms.

We have added text to discuss more on this. | *(See Section 4. Discussion Paragraph 4, Line 27 of the revised manuscript).*
In spite of the challenges associated with varying geo-spatial resolution of remotely sensed imagery, including decreased chances to detect plastic litter in the visible spectrum (Acuña-Ruz et al., 2018), satellites provide essential information about the environment. Satellite missions with TIR sensors include ASTER from the National Aeronautics and Space Administration/Japanese Ministry of Economy Trade and Industry, ECOSTRESS from National Aeronautics and Space Administration as well as Landsat-8 from the United States Geologic Survey. The capabilities (TIR spectral, geo-spatial, revisit interval) of ASTER, ECOSTRESS and Landsat-8 missions must be assessed with a focus on detecting aggregated litter zones, considering the geo-spatial resolutions of these sensors (38 - 100 m). We need to further emphasize that the atmospheric window in the TIR is relatively wide. This atmospheric window contained a significant number of diagnostic wavebands of anthropogenic materials we studied and it would be vital to explore development of detection algorithms using the limited (2 - 5 wavebands) spectral information available on these current TIR missions. |
| **C5.**
The ECOSTRESS spectral library includes a number of manmade and natural materials. I don't know if it includes materials similar to the samples collected and measured by the authors. In this case, a rough comparison between these measurements could be an option for a brief test of the data presented by the authors. | **R5.**
We checked the ECOSTRESS repository and the similarities are based on the material being anthropogenic or synthetic products. We think it does merit a comprehensive comparison of these datasets but it would fall out of the scope of the current manuscript.

We acknowledge this point in the discussion section and highlight the some related libraries/studies. | *(See Section 4. Discussion Paragraph 1, Line 8 of the revised manuscript).*
We are convinced our TIR sample subset provide invaluable complementary insights to the interdisciplinary scientific evidence-based knowledge global plastic litter. To this end, it is recommended that within the TIR remote sensing community a comprehensive high quality assured and quality controlled spectral reference library be established to carefully harmonize available TIR measurements from various works e.g. ECOSTRESS (Meerdink et al., 2019), SLUM (Kotthaus et al., 2014) or contaminated anthropogenic surfaces (Kerekes et al., 2008). |

**References**

Acuña-Ruz, T., Uribe, D., Taylor, R., Amézquita, L., Guzmán, M. C., Merrill, J., Martínez, P., Voisin, L., and Mattar B., C.: Anthropogenic marine debris over beaches: Spectral characterization for remote sensing applications, Remote Sens. Environ., 217, 309-322, doi:10.1016/j.rse.2018.08.008, 2018.

Kerekes, J. P., Strackerjan, K.-E., and Salvaggio, C.: Spectral reflectance and emissivity of man-made surfaces contaminated with environmental effects, Opt. Eng., 47, 106201(106201-106210), doi:10.1117/1.3000433, 2008.

Kotthaus, S., Smith, T. E. L., Wooster, M. J., and Grimmond, C. S. B.: Derivation of an urban materials spectral library through emittance and reflectance spectroscopy, ISPRS J. Photogramm. and Remote Sens., 94, 194-212, doi:10.1016/j.isprsjprs.2014.05.005, 2014.

Meerdink, S. K., Hook, S. J., Roberts, D. A., and Abbott, E. A.: The ECOSTRESS spectral library version 1.0, Remote Sens. Environ., 230, 111196, doi:10.1016/j.rse.2019.05.015, 2019.

Schodlok, M. C., Whitbourn, L., Huntington, J., Mason, P., Green, A., Berman, M., Coward, D., Connor, P., Wright, W., Jolivet, M., and Martinez, R.: HyLogger-3, a visible to shortwave and thermal infrared reflectance spectrometer system for drill core logging: functional description, Aust. J. Earth Sci., 63, 929-940, doi:10.1080/08120099.2016.1231133, 2016.

---

## Author Comment (AC3) · 3 Sep 2020

**Reply to reviewer**

**Manuscript Title** : Hyperspectral longwave infrared reflectance spectra of dry anthropogenic plastics and natural materials
**Authors** : Garaba, Acuña-Ruz and B. Mattar
**Journal** : *Earth System Science Data* (*ESSD*)

**Anonymous Referee #3**

| Comment | Response | Revision Implemented |
|---|---|---|
| **C1**. This manuscript describes a dataset of reflectance/emissivity spectra of dry manmade and natural materials from 6000 nm to 14000 nm. The dataset has the potential to be very useful for identification of litter in marine and coastal environments as there doesn't seem any other plastic spectral libraries available (nor even plastics in other spectral libraries such as the ECOSTRESS spectral library) and research into marine and plastic pollution is clearly gaining interest and awareness.

In general, I think the manuscript is organised well and a promising accompaniment to the dataset. However, I think a few revisions are needed before this should be published, primarily to the methodology section in order to enable maximum clarity for users. I've gone into some details about what I think should be addressed – apologies for the length but I think it's because the dataset has the potential to be really useful. Since this is to accompany a dataset, it needs to be very clear for the reader how the samples were collected, prepared and measured. I think it's clear from the manuscript how the samples were collected but there is limited information about preparation and further clarity is required about the sample measurement, particularly since there isn't any data available on accuracy of the HyLogger. | **R1**. We thank the reviewer for the detailed comments and constructive suggestions on our manuscript. | None |

| Comment | Response | Revision Implemented |
|---|---|---|
| **C2.**
Questions I might want to know if I were to use this dataset which are absent from the methodology include:

i. How long was there between collection and measurement?

ii. How and why were there samples dried? | **R2.**
These are key details you highlight. In summary, samples after collection were characterized, labelled, geo-referenced and carefully moved inside sealed boxes for further laboratory analyses. It took about a week from field collection to laboratory measurements. We tried to best simulate the environment we obtained the samples hence the drying procedure.

We have added these points to the manuscript and the title has been revised to make this clear. | *(See Section 2.2 Directional hemispherical reflectance measurements Page 4, Line 12 of the revised manuscript).*
Before the spectral measurements, the algae was placed between newspapers to dry whilst the rest were left to dry naturally in the laboratory for 7 days, this step was conducted to best simulated the conditions from which the litter originated along the shoreline, a relatively dry environment exposed to wind gusts and sunlight. |
| **C3.**
This is important as surface moisture has been shown to impact surface reflectance in the LWIR (e.g. see https://doi.org/10.1016/j.rse.2010.02.002). Surely wet samples might be more representative of the conditions you'd see in marine environments? | **R3.**
We agree surface moisture can influence reflectance measurements in the LWIR. It is crucial that follow up research should assess these effects.

We added a line to emphasize this point and have added the reference suggested reference. | *(See Section 4. Discussion Page 15, Line 10 of the revised manuscript).*
During our experiments the samples were left to dry naturally but future studies are urged to investigate wet samples to better simulate aquatic floating material as surface moisture has been shown to affect the detectable signal in the SWIR (Garaba and Dierssen, 2018) as well as in LWIR (Hulley et al., 2010). |
| **C4.**
iii. What is this 'inside' and 'outside' that is mentioned in the results – did you cut into the samples to measure the 'inside'? You need to describe this since the impact this has on reflectance/emissivity is non-negligible. | **R4.**
We understand that it was not clear and have added text to highlight it was meant to qualitatively distinguish the surfaces as shiny/brighter or dull/weathered. | *(See Section 2.1 Samples Page 3, Line 2 of the revised manuscript).*
Visual inspection of litter samples suggested short to long term exposure to natural weathering processes in the environment. Several objects seemed to have significant apparent variations in colour or brightness, we therefore completed measurements on the respective surfaces to assess the effects of these differences on the reflectance (**Figure 1b, c, g**). For brevity, the surfaces were identified as inside and outside for these individual objects and no cutting or other preparations were done on these materials. |
| **C5.**
iv. Was a background radiance measurement made as detailed in Schodlok et al 2016? | **R5.**
Yes, the Hylogger 3 system includes an auto calibration procedure to heat up and collect background as well as spectral and infragold measurements.

Information has been appended to highlight this step. | *(See Section 2.2 Directional hemispherical reflectance measurements Page 4, Line 10 of the revised manuscript).*
Detailed specifications of the instrument have been reported in a prior study and we conducted our experiments following the proposed operating protocol of the instrument (Schodlok et al., 2016). |

| Comment | Response | Revision Implemented |
|---|---|---|
| **C6.**
v. Is there any information about signal-to-noise for the instrument and the measurements? | **R6.**
Yes, it is ≥ 2000 at 8 µm at peak signal-to-noise ratio for a Lambertian material with 100 % directional hemispherical spectral reflectance. | *(See Section 2.2 Directional hemispherical reflectance measurements Page 4, Line 8 of the revised manuscript).*
HyLogger-3™ spectrometer has 341 wavebands and a peak signal-to-noise ratio (≥2000 at 8µm) for a Lambertian surface with 100 % directional hemispherical spectral reflectance. |
| **C7.**
vi. You discuss spectra being grouped into associated materials (l.27, p.4) – what are these group and how were these groups determined? Do you just mean e.g. all sands, all styrofoams? | **R7.**
The grouping was based on the visual inspection of the objects and the spectral characteristics determined. Sands referred to all sand samples and Styrofoam represented all similar materials.

A sentence has been added to further explain this. | *(See Section 2.3. Data Analyses Page 5, Line 25 of the revised manuscript).*
The classification of these materials was therefore based on spectral characteristics revealed by the statistical analyses complemented by careful visual inspection of the objects (**Figure 1**). |
| **C8.**
vii. The authors refer to 'length' in line 10, p.4. What does this refer to, length of the tray or length of time? | **R8.**
We meant the inherent length of each object.

It has been revised to elaborate on this point. | *(See Section 2.2 Directional hemispherical reflectance measurements Page 5, Line 5 of the revised manuscript).*
Number of scans per object ranged between 12 to 99 scans and this was automatically set by the instrument as a function of the inherent length of each item along the track of scanning. |
| **C9.**
viii. Where were the measurements made? Were the samples sent to CSRIO Australia for measurement on the setup detailed in Schodlok et al (2016) or is there a setup in Chile where they were measured? If measurements were made using a different setup to the one in Schodlok et al (2016), I would suggest you include some more information about it and perhaps an example image of the setup during a measurement for the user. You could also perhaps could include table to present number of scans by sample/tray of samples which would be useful for the user. | **R9.**
Spectral measurement were completed at the University of Chile and the standard operating protocol was consistent with the study of Schodlok et al. 2016. As the scans were automated we feel it is not very useful to the reader. However, if the reviewer still thinks it would be key information we are glad to provide the information.

We have added additional information about the protocol and location of instrument. | *(See Section 2.2 Directional hemispherical reflectance measurements Page 4, Line 7 of the revised manuscript).*
Thermal infrared spectral measurements between 6 and 14.5 µm were obtained in 0.025 µm steps using the laboratory hyperspectral HyLogger-3™ spectrometer at the University of Chile, Chile. HyLogger-3™ spectrometer has 341 wavebands and a peak signal-to-noise ratio (≥2000 at 8µm) for a Lambertian surface with 100 % directional hemispherical spectral reflectance. Detailed specifications of the instrument have been reported in a prior study and we conducted our experiments following the proposed operating protocol of the instrument (Schodlok et al., 2016). |

| Comment | Response | Revision Implemented |
|---|---|---|
| **C10.**
ix. How did you get the spectra from the HyLogger imagery? Are your spectra the average of multiple spatial pixels? | **R10.**
As the statistical average value of all the "pixels" captured by the "line scan" over the samples. In this regard, each scan was a pixel.

We now indicate this in the manuscript. | *(See Section 2.2 Directional hemispherical reflectance measurements Page 5, Line 7 of the revised manuscript).*
Altogether, 76 spectra matching the number of objects in this study were computed as the average of successive scans over each respective item. A true colour image was also captured during the scanning of each subset placed on the tray. |
| **C11.**
If the samples and measurement protocol presented in this paper are the same considered in Acuña-Ruz et al. (2018), the authors could answer some of the above simply by referencing that. However, I'm not sure they are since this paper talks about 76 samples while the Acuña-Ruz et al (2018) paper talks about over 144 samples. | **R11.**
The sampling protocol and materials were from Acuña-Ruz et al. (2018).

We have referenced the study as suggested. | *(See Section 2.1 Samples Page 2 Line 31 of the revised manuscript ).*
Litter was gathered along the shorelines of Punta Mallil-Cuem, Detif and Punta Apabón on Chiloé Archipelago, Los Lagos region of Chile from January to February 2017 (Acuña-Ruz et al., 2018). |
| **C12.**
In terms of accompanying figures and tables, generally these are good although I think a few of the figures could benefit from further explanation in the captions. In Figure 1 for example, I think the sample key needs to explained in more detail in the caption. | **R12.**
We have revised the caption. | *(See Figure 1 Caption Section 2.1 Samples of the revised manuscript).*
Figure 1. Naturally dried (a) sands and shells, (b-c) Styrofoam® outside and inside surfaces, (d) nautical ropes, construction material, gunny sacks and fish nets, (e) tubes, plastic bottles and buoys, (f) gunny sacks, construction material, meshes, fish nets and algae; and (g) buoys pieces collected along the shorelines of Punta Mallil-Cuem, Detif and Punta Apabón on Chiloé Archipelago, Chile from January to February 2017 placed on black trays for hyperspectral hemispherical reflectance measurements using the HyLogger-3™ spectrometer. |
| **C13.**
Also, which of the pictured repeats for N37, N44 and N47 are the 'inside' and 'outside'? | **R13.**
We have appended text to indicate it was meant to qualitatively distinguish the surfaces as shiny/brighter or dull/weathered. Please see our response **R4.** | See our response **R4** and the corresponding revisions. |

| Comment | Response | Revision Implemented |
|---|---|---|
| **C14.**
The results and discussion are in general well presented with good consistency for each subsection in the results. A couple of points I had here: - Is the end-member presented in each subsection the mean spectra of multiple scans? Unclear from the text | **R14.**
It was the mean spectra derived from those presented in each respective (b) subplot.

We include a sentence to clarify this point. | *(See Section 2.3. Data Analyses Page 5, Line 5 of the revised manuscript).*
Representative end-members were estimated as the average of all spectra in each proposed group. |
| **C15.**
- I think you would benefit from further discussion of UPD and variability as it's not clear why you have considered this nor how you have used it. If you're using it to be a measure of how trustworthy the spectra is (as I think you are?), a comparison of the different UPDs would be useful to see in the discussion | **R15.**
UPD was used as a trustworthy indicator and we have added further text in the discussion to evaluate its importance.

A discussion has been added to explain the relevance of the metric. | *(See Section 4. Discussion Page 14, Line 21 of the revised manuscript).*
The UPD metric we investigated showed how large or small the magnitude differed in our proposed groups. In general, we observed moderate percentage differences, suggesting algorithm development based on the magnitude of reflectance might have large uncertainties. Alternatively, band ratioing algorithms or spectral shape based algorithms would mitigate the problems linked to variations in magnitude. |
| **C16.**
- I don't think Figure 10 is necessary | **R16.**
We agree with the reviewer. | Figure 10 has been deleted. |

| Comment | Response | Revision Implemented |
|---|---|---|
| **C17.**
- I was surprised to see no discussion of other spectral libraries (e.g. ECOSTRESS spectral library, SLUM spectral library) in the introduction and/or discussion given that this dataset will have a complementary role to these. If possible I would suggest you show an inter-comparison with data from these spectral libraries or other papers to help the user understand the comparable performance of your dataset since you don't have calibration or accuracy information for the HyLogger. This is especially important as you are observing unrealistic negative reflectances which could suggest inaccurate measurements. As noted earlier however, I couldn't find any plastics in the ECOSTRESS spectral library so you'd probably have to do this comparison with the sand, styrofoam or algae samples if you could find similar samples. | **R17.**
We agree with the reviewer that there is a need for inter-comparison of datasets. We think it is out of the scope of our current manuscript but we acknowledge the need to harmonize different spectral libraries.

We have referenced examples of libraries with anthropogenic material for future efforts to harmonize such datasets. Although the material might not be similar the variability in the sampled materials add value to the spectral libraries in open-access.

In the case of unrealistic negative reflectances, we carefully re-checked the data set. We confirmed that the algae and the fishnet (F2) had anomalous values suggesting possible detection limits of sensor hence noise was recorded by detector. We also noticed that the net was very degraded, thus when a background correction was applied to both the algae and net we ended up with negative values. It does merit further investigations to better understand this finding. | *(See Section 4. Discussion Page 14, Line 8 of the revised manuscript).*
We are convinced our TIR sample subset provides invaluable complementary insights to the interdisciplinary scientific evidence-based knowledge of global plastic litter. To this end, it is recommended that within the TIR remote sensing community a comprehensive high quality assured and quality controlled spectral reference library be established to carefully harmonize available TIR measurements from various works e.g. ECOSTRESS (Meerdink et al., 2019), SLUM (Kotthaus et al., 2014) or contaminated anthropogenic surfaces (Kerekes et al., 2008). |
| **C18.**
In terms of usefulness of this dataset, there are two points I wanted to make:
1) You identify in your discussion that a limitation of your dataset is that you don't have information on the chemical composition of your samples. However, I think using terms like 'other plastics' is very vague and will limit the use in applications – could you be at all more specific? For example, 'other plastics' seems to have multiple absorption lines, which one will users know to use? Also, in the accompanying sample pictures, are these the 'buoy samples' and 'buoy2_samples'? | **R18.**
We understand indicating 'other plastics' might have been a vague description. The idea was to classify the rest of the material in a concise manner that was also consistent with matching spectral shapes. The "other plastics" class consisted of pieces of plastic PET (polyethylene terephthalate) bottles, nets are used for aquaculture productive systems, foam, black construction material and strap piece.

We have add text to improve the description of the items by including a description of the material and we have also included this information in the caption of Figure 9, Table 1 and section 3.2.5 heading | *(See Section 3.2.5, Heading and Caption Figure 9, Table 1 of the revised manuscript).*

**3.2.5 Plastic bottles, nets, foam and other plastics**

**Figure 9.** Spectral reflectance of other anthropogenic materials (a) green plastic bottle pieces and transparent strip, (b) mesh and fishing net, (c) black construction materials, (d) gunny sack like patch, (e) polyurethane foam piece and (f) nautical cargo strap in **Figure 1** gathered along the shorelines of Punta Mallil-Cuem, Detif and Punta Apabón on Chiloé Archipelago, Chile. An end-member spectrum is proposed with diagnostic absorption features highlighted by the vertical lines. |

| Comment | Response | Revision Implemented |
|---|---|---|
| **C19.**
2) If you're advising the user that this dataset can be used with TIR satellite sensors, you really need to address the issue of spatial and spectral resolution. Would ASTER or Landsat 8's spatial resolution really be high enough to detect samples of this kind? Even the highest resolution TIR sensors (ECOSTRESS, HyspIRI e.g.) have spatial resolutions of 60m + and with SLSTR you're looking at 1 km. If you're going to argue that this dataset can be used for satellite sensors, you'll need something similar to the discussion in e.g https://doi.org/10.1038/s41598-020-62298-z to show suitability of thermal sensors for plastic detection in oceans (and therefore why spectral library is required). If the plastic observed is < 60m, I would advise instead moving the introduction and discussion a bit more towards hyperspectral airborne TIR remote sensing (e.g. using NASA's HyTES, Specim's OWL, TASI) and thermal UAVs. Use of hyperspectral airborne sensors has the benefit of avoiding the issue of absorption features being outside satellite spectral bands. | **R19.**
Thank you for raising this point. Our aim here was to discuss the prospects of current mission and the need for synergy in instrumentation available relevant to plastics.

We now discuss the geo-spatial and spectral capabilities of the current satellite missions (ASTER, ECOSTRESS and Landsat 8).

We also highlight the prospects of the airborne platform as suggested. | *(See Section 4. Discussion Page 15, Line 24 of the revised manuscript).*
The benefit of conducting aerial field surveys using multi to hyperspectral sensors (e.g. NASA HyTES, Specim OWL, ITRES TASI-600 or SensyTech AHS) on aircrafts will be the possibility of capturing high geo-spatial TIR imagery in near-similar approaches/conditions to those of satellites i.e. an intervening atmosphere and a mobile platform. In spite of the challenges associated with varying geo-spatial resolution of remotely sensed imagery, including decreased chances to detect plastic litter in the visible spectrum (Acuña-Ruz et al., 2018), satellites provide essential information about the environment. Satellite missions with TIR sensors include ASTER from the National Aeronautics and Space Administration/Japanese Ministry of Economy Trade and Industry, ECOSTRESS from National Aeronautics and Space Administration as well as Landsat-8 from the United States Geologic Survey. The capabilities (TIR spectral, geo-spatial, revisit interval) of ASTER, ECOSTRESS and Landsat-8 missions must be assessed with a focus on detecting aggregated litter zones, considering the geo-spatial resolutions of these sensors (38 - 100 m). We need to further emphasize that the atmospheric window in the TIR is relatively wide. This atmospheric window contained a significant number of diagnostic wavebands of anthropogenic materials we studied and it would be vital to explore development of detection algorithms using the limited (2 - 5 wavebands) spectral information available on these current TIR missions. |
| **C20.**
Regarding the dataset itself, it's accessible and easy to use (although note that the KML file is not mentioned in the accompanying publication). You could consider separating the metadata and the data for ease of use. I would also advise including a key with the sample images. | **R20.**
We are in the process of incorporating the suggested edits to the Pangaea dataset. | *A KML file and additional images were attached published on Pangaea see the Further Details section of the dataset.* |

| Comment | Response | Revision Implemented |
|---|---|---|
| **C21.**
The abstract here could benefit from copy-editing. | **R21.**
We asked a native speaker to carefully provide copy-editing of the manuscript. | The manuscript was proof read by a native-speaker. |
| **C22.**
Finally, the manuscript was in general well-written but there are a few typos and incomplete sentences in the manuscript that suggest the need for a copy edit. A few I noticed in the manuscript:
1. p.14 line 7 has missing end to sentence | **R22.**
Revised. | *(See Section 4. Discussion Page 15 Line 27 of the revised manuscript).*
In spite of the challenges associated with varying geo-spatial resolution of remotely sensed imagery, including decreased chances to detect plastic litter in the visible spectrum (Acuña-Ruz et al., 2018), satellites provide essential information about the environment. Satellite missions with TIR sensors include ASTER from the National Aeronautics and Space Administration/Japanese Ministry of Economy Trade and Industry, ECOSTRESS from National Aeronautics and Space Administration as well as Landsat-8 from the United States Geologic Survey. |
| **C23.**
2. line 7 on p. 3 incorrectly says 'Were believe' | **R23.**
Revised. | *(See Section 2.1 Samples Page 3, Line 10 of the revised manuscript).*
The samples collected for this experiment were assumed to represent a majority of anthropogenic plastic and natural materials found along the shorelines of Chiloé Archipelago and this was consistent with floating litter obtained from multi-year surveys of other regions in Chile (Thiel et al., 2013;Urbina et al., 2020). |
| **C24.**
3. line 15 p.6, should this be 12000 nm rather than 1200 nm? | **R24.**
Thank you for pointing this out. It should be 12 μm. | *(See Section 3.1.3 Algae Page 7, Line 15 of the revised manuscript).*
10.6 and 12 μm (**Figure 4**). |
| **C25.**
4. The sentence commencing l.23 on p.4: 'An inter-comparison of: : :' needs to be rewritten | **R25.**
The sentence has been revised. | *(See Section 3.1.3 Algae Page 7, Line 15 of the revised manuscript).*
The waveband locations of diagnostic absorption features were first obtained from the modified scale-space peak algorithm and further confirmed through major peaks revealed in second derivative spectra. |

| Comment | Response | Revision Implemented |
|---|---|---|
| **C26.**
Also, a very minor point but I would consider changing the units from nanometre to micrometre throughout as the thermal infrared spectroscopy community tends to use microns more. | **R26.**
We agree and have revised the x-axis of all the figures and in the manuscript as suggested. | *(See the x-axis of the revised Figure 2 to Figure 9 of the revised manuscript)* |

**References**

Acuña-Ruz, T., Uribe, D., Taylor, R., Amézquita, L., Guzmán, M. C., Merrill, J., Martínez, P., Voisin, L., and Mattar B., C.: Anthropogenic marine debris over beaches: Spectral characterization for remote sensing applications, Remote Sens. Environ., 217, 309-322, doi:10.1016/j.rse.2018.08.008, 2018.

Garaba, S. P., and Dierssen, H. M.: An airborne remote sensing case study of synthetic hydrocarbon detection using short wave infrared absorption features identified from marine-harvested macro- and microplastics, Remote Sens. Environ., 205, 224-235, doi:10.1016/j.rse.2017.11.023, 2018.

Hulley, G. C., Hook, S. J., and Baldridge, A. M.: Investigating the effects of soil moisture on thermal infrared land surface temperature and emissivity using satellite retrievals and laboratory measurements, Remote Sens. Environ., 114, 1480-1493, doi:10.1016/j.rse.2010.02.002, 2010.

Kerekes, J. P., Strackerjan, K.-E., and Salvaggio, C.: Spectral reflectance and emissivity of man-made surfaces contaminated with environmental effects, Opt. Eng., 47, 106201(106201-106210), doi:10.1117/1.3000433, 2008.

Kotthaus, S., Smith, T. E. L., Wooster, M. J., and Grimmond, C. S. B.: Derivation of an urban materials spectral library through emittance and reflectance spectroscopy, ISPRS J. Photogramm. and Remote Sens., 94, 194-212, doi:10.1016/j.isprsjprs.2014.05.005, 2014.

Meerdink, S. K., Hook, S. J., Roberts, D. A., and Abbott, E. A.: The ECOSTRESS spectral library version 1.0, Remote Sens. Environ., 230, 111196, doi:10.1016/j.rse.2019.05.015, 2019.

Schodlok, M. C., Whitbourn, L., Huntington, J., Mason, P., Green, A., Berman, M., Coward, D., Connor, P., Wright, W., Jolivet, M., and Martinez, R.: HyLogger-3, a visible to shortwave and thermal infrared reflectance spectrometer system for drill core logging: functional description, Aust. J. Earth Sci., 63, 929-940, doi:10.1080/08120099.2016.1231133, 2016.

Thiel, M., Hinojosa, I. A., Miranda, L., Pantoja, J. F., Rivadeneira, M. M., and Vásquez, N.: Anthropogenic marine debris in the coastal environment: A multi-year comparison between coastal waters and local shores, Mar. Pollut. Bull., 71, 307-316, doi:10.1016/j.marpolbul.2013.01.005, 2013.

Urbina, M. A., Luna-Jorquera, G., Thiel, M., Acuña-Ruz, T., Amenábar Cristi, M. A., Andrade, C., Ahrendt, C., Castillo, C., Chevallier, A., Cornejo-D'Ottone, M., Correa-Araneda, F., Duarte, C., Fernández, C., Galbán-Malagón, C., Godoy, C., González-Aravena, M., I.A., H., Jorquera, A., Kiessling, T., Lardies, M. A., Lenzi, J., C., M. B., Munizaga, M., Olguín-Campillay, N., Perez-Venegas, D. J., Portflitt-Toro, M., Pozo, K., Pulgar, J., and Vargas, E.: A country's response to tackling plastic pollution in aquatic ecosystems: The Chilean way, Aquat. Conserv.: Mar. Freshw. Ecosyst., doi:10.1002/aqc.3469, 2020.